# Exciting and Detecting Higher-Order Guided Lamb Wave Modes in High-Density Polyethylene Structures Using Ultrasonic Methods

**DOI:** 10.3390/ma17010163

**Published:** 2023-12-28

**Authors:** Justina Šeštokė, Elena Jasiūnienė, Reimondas Šliteris, Renaldas Raišutis

**Affiliations:** 1Prof. K. Barsauskas Ultrasound Research Institute, Kaunas University of Technology, LT-51423 Kaunas, Lithuania; elena.jasiuniene@ktu.lt (E.J.); reimondas.sliteris@ktu.lt (R.Š.); renaldas.raisutis@ktu.lt (R.R.); 2Department of Electronics Engineering, Kaunas University of Technology, Studentu Str. 50, LT-51368 Kaunas, Lithuania

**Keywords:** material characterization, polyethylene structures, higher-order guided waves, finite element modelling, ultrasonic methods

## Abstract

High-density polyethylene (HDPE) pipes are becoming increasingly popular, being used in various fields, such as construction, marine, petroleum, water transfer, process water, methane gas collection, oil and gas gathering, gas distribution systems, mining, acid and wet gas lines, offshore oil and gas and in nuclear power plants. Higher-order guided Lamb wave (UGW) modes can be used to detect various defects in complex structures. We will apply this methodology to one of the types of plastic—the structure of high-density polyethylene (HDPE). However, the excitation of UGW modes faces numerous challenges, especially when there is a need to identify which mode is excited. It is essential to note that, in the higher frequency range, multiple different higher-order modes can usually be excited. This can make it difficult to determine which modes have actually been excited. The objective of this research was to successfully excite and receive various higher-order UGW modes in high-density polyethylene structures using both ultrasonic single-element transducers and a phased array. Theoretical calculations were performed using a variety of methods: semi-analytical finite element (SAFE) method, 2D spatial–temporal spectrum analysis and finite element modeling (FEM). The results obtained from both measurements and simulations clearly demonstrate the possibility of efficiently exciting and receiving different Lamb wave modes possessing different phase velocities.

## 1. Introduction

HDPE (high-density polyethylene) is one of the most stable and economical thermoplastics on the market today [1]. The value of the global HDPE market exceeds 63 billion per year [2]. Because HDPE is a versatile material, there are a number of uses for it. Because of their chemical properties, HDPE materials are resistant to temperature, pollution, moisture, scratches and dents and are highly resistant to corrosion. Various and durable constructions are made from this material, which are used as storage for agrochemicals and liquids such as water, gas and oil. HDPE is weather-resistant in most cases. It can withstand extreme temperatures (−130 to 80 °C degrees), making it a strong insulator and preventing it from cracking or bursting in freezing conditions [3,4,5,6]. In the chemical industry, it is imperative that systems for critical processes need to be safe and reliable. The fluids involved in these processes are not only frequently expensive, but also highly toxic, posing potential environmental hazards. Therefore, it is necessary to check and control to avoid leakage of chemicals or gases, to avoid damage to the environment, human health, and large losses. By carrying out the planned diagnostics of structures, faults can be eliminated in time and easily. In this way, catastrophic failures of devices/structures, which can strongly affect the natural environment, can be avoided. Previous and current diagnostic methods for plastic pipes include different NDT (non-destructive testing) techniques such as visual inspection, acoustic emission, laser scanning or radiography [7,8,9,10].

When scheduled diagnostics of a device/structure is performed, there is a high probability of finding and identifying a defect in the device/structure at an early stage. Timely detection of defects prevents complicated troubleshooting. In order to maintain the safety of devices/structures, effective and economical diagnostic methods are needed [11,12,13,14,15,16,17]. In the literature reviewed, most of the methods used for diagnosing defects were applied to the steel pipes. Acoustic emission, radiography, ground-penetrating radar, visual inspection and thermal imaging were used [18,19,20,21,22,23,24,25]. Very little diagnostic work has been done on polyethylene-type materials, i.e., Lowe et al. investigated the second-order longitudinal ultrasonic guided waves (UGW) mode in HDPE pipes [26]. Today, one of the types of polyethylene, as a high-density plastic which has particularly good chemical and mechanical resistance, is used in pipelines for water, oil and gas. It is necessary to develop diagnostic methods for detecting different types of defects in this material.

Typically, the lowest zero-order antisymmetric A_0_ Lamb waves are extensively used for non-destructive testing due to their sensitivity to different defects, which are particularly common in composite materials. However, in samples with a non-sheet structure, higher-order symmetric and antisymmetric Lamb waves could be used, which have very different properties from the lowest-order Lamb modes. Higher-order Lamb waves can propagate at higher frequencies than zero-order symmetric and antisymmetric waves. They are generally more diffuse compared to zero-order waves. In the frequency ranges above the defined particular cutoff frequencies of the zero-order modes, there are always not a single mode but several higher order modes [27]. This, however, complicates the analysis of received ultrasound signals. The spatial distribution of the displacements of higher-order modes in waveguides differs significantly from that of zero-order modes. This means that their interactions with non-homogeneities inside the material can provide much more detailed information about them than a single zero-order antisymmetric mode alone. To date, the majority of publications have focused on analyzing the application of the zero-order and lower-frequency antisymmetric mode for defect detection in various composite and non-composite materials [28,29]. The applications of higher-order modes are limited due to their more complex excitation and reception. Ultrasonic phased arrays can be adapted for this purpose, but they are mainly used for excitation and reception of the guided wave propagating at a lower frequency.

This article studies higher-order ultrasonic waves using contact ultrasonic techniques, with a particular focus on the use of piezoelectric transducers. Their widespread availability and low cost make them appealing for a wide range of applications.

The objective of this investigation was to efficiently excite and detect various higher-order guided Lamb wave modes in high-density polyethylene structures, employing single-element ultrasonic transducers and a phased array. Theoretical calculations were performed using different methods: the semi-analytical finite element (SAFE) method, 2D spatial–temporal spectrum analysis and finite element modeling (FEM).

The paper consists of six sections. In Section 2, the properties of higher-order guided wave modes in high-density polyethylene structures are presented. The experimental investigation using the ultrasonic method is described in Section 3. In Section 4, the theoretical investigation using finite element modeling (FEM) is presented. The discussion of the results obtained is given in Section 5, and the conclusions are presented in Section 6.

## 2. Object of Investigation

The material investigated in this work was high-density polyethylene HDPE PE-300 (NSB Polymers GmbH, Dormagen, Germany). As mentioned earlier, it is one of the most sought-after materials for liquid containers. The pipes are becoming increasingly popular and can be used in a various applications including construction, marine, oil, water transfer, process water, methane gas collection, oil and gas gathering, gas distribution systems, mining, sour and wet gas lines, offshore oil and gas, and have particular relevance in nuclear power plants. After reviewing different types of plastics, it can be concluded that HDPE has significantly lower mechanical and thermal properties, which can be defined as bending, tensile stress or compressive strength. If we judge by high-molecular-weight PE, then the plastic we offer for research is more rigid; for this reason, HDPE is resistant to continuous impact. The purpose of this section is to describe the properties of the selected material. This is the basis for measuring the higher-order modes in the selected specimen. Our previous work described exactly how the properties of this material were measured. The measurements were performed at the Department of Physical Acoustics (Bordeaux, France). Measurements were performed on samples of various thicknesses, but in this work, only a 20 mm thick HDPE sample was investigated. The HDPE PE-300 sample is an isometric material, possessing elastic properties *c*_22_ = *c*_11_, *c*_33_ = *c*_11_, *c*_44_ = *c*_66_, *c*_55_ = *c*_66_, *c*_12_ = *c*_11_ − 2*c*_66_, *c*_13_ = *c*_12_, *c*_23_ = *c*_12_. The properties of the specimens of different thicknesses were the same, and the elastic parameters of the HDPE PE-300 specimen are presented in Table 1.

The velocities of longitudinal and transverse waves are independent of frequency. The ultrasonic longitudinal *c_L_* and transverse *c_T_* wave velocities are calculated using the formulas Equations (1) and (2) [30].
(1)$$c_{L}=\sqrt{\frac{c_{11}}{\rho }}=3010\;m/s$$
(2)$$c_{T}=\sqrt{\frac{c_{66}}{\rho }}=743\;m/s$$

First of all, we performed theoretical calculations of the selected sample. The calculations of the Lamb waves were performed using the semi-analytical finite element (SAFE) method [27]. The calculated dispersion curves of the selected specimen (thickness of 20 mm) are presented in Figure 1a (the phase velocity) and Figure 1b (the group velocity).

One of the aims of this work is to excite a higher-order mode using a low-frequency ultrasonic phased array. It can be seen from the dispersion curves that the higher-order A_1_ mode can be excited from the frequency of 30 kHz. To verify the simulation results, experimental measurements were performed. The following Section 3 presents an experimental measurement using ultrasonic transducers to obtain dispersion curves.

## 3. Experimental Investigation Using Ultrasonic Method

This section presents an experimental investigation using ultrasonic methods. The aim of the whole investigation is to compare results of the numerical calculation method with the experimental one. The determined or calculated dispersion curves align with the measured ones. The measurements were made using ultrasonic transducers that are contact type. Selected ultrasonic transducers operated within the low-frequency band (50–350 kHz). Ultrasonic transducers manufactured at the Ultrasound Institute (UI) of the Kaunas University of Technology (Kaunas, Lithuania). These transducers have a 5 mm diameter convex shield and a contact area diameter of 0.5 mm. For enhanced stable acoustic contact with the tested sample, glycerol as contact fluid was used for the receiver and the transmitter. This liquid provides a more stable acoustic contact, and then a higher accuracy of measurements is obtained, because the acquired ultrasonic signals are more stable. The ultrasonic equipment used for the experiment is shown in Figure 2. It consists of an amplifier (UI, Kaunas, Lithuania), mentioned contact ultrasonic transducers with a frequency of *f* = 140 kHz, and an ULTRALAB ultrasonic measurement system (UI, Kaunas, Lithuania), which was connected together with a linear scanner, the latter manufactured by the Ultrasound Institute (UI) of the Kaunas University of Technology (Kaunas, Lithuania) [31,32,33,34]. To excite higher-order antisymmetric modes (A_3_ and A_4_), the excitation frequency was set to *f* = 140 kHz. The frequency of the transmitter transducer was adjusted until the maximum signal amplitude was obtained. The sampling frequency was 100 MHz. For experimental investigation, an HDPE PE-300 specimen (thickness of 20 mm) and dimensions of 700 × 700 mm was used. The scanning step was 0.1 mm, and the scanning distance from the transmitter was 150 mm. The distance between the contact transducers (transmitter and receiver) was 90 mm. This distance was the initial position of the receiver from the transmitter.

The experimental measurement was performed using an excitation signal with burst of three periods and voltage amplitude of 300 V. Figure 3 shows the excitation signal that was excited by an ultrasonic contact transducer to the HDPE PE-300 sample plate. Please note that the signal amplitude is given in arbitrary units (n.u.). A B-scan image of the selected specimen is shown in Figure 4.

The B-scan image acquired shows the excitation of higher-order antisymmetric modes including A_3_ and A_4_. The SAFE simulation method validates measurement results obtained using the contact excitation.

First, 2D spatial–temporal spectrum analysis of normal displacement B-scans was used to determine the dispersion curves of the test specimen [32]. This method is extensively described in our previous work, so it is not described here. Figure 5 shows the calculation and measurement results. The calculation results, depicted by red lines, were obtained using the SAFE method. The same figure displays the measurement results obtained using the 2D FFT (Fast Fourier Transform) in the B-scan, frequency-converted phase-velocity domain. It is important to note that the amplitudes are given in arbitrary units and the amplitude scale is depicted on the right side of the figure. Figure 5b presents the results using the particular signal acquired at the 75 mm distance of the measured propagating modes within the B-scan image for the HDPE PE-300 sample, where symmetric modes are represented by red lines, antisymmetric modes by blues lines and shear modes by green lines. Figure 5c presents phase velocities of antisymmetric and symmetric modes up to 170 kHz, and the A_3_ mode pattern is visible at 140 kHz.

A sufficient agreement was obtained between the SAFE method and the measured-as-well-as-processed 2D FFT results. These findings confirm the capability to excite anti-symmetric A_0_, A_1_, A_2_, A_3_, A_4_, A_5_, A_6_ and symmetric S_0_, S_1,_ S_2_, S_3_, S_4_, S_5_, S_6_ modes using the ultrasonic contact-type transducers (operation frequency range from 40 kHz up to 250 kHz) (Figure 5). The obtained results further indicate that the highest displacement amplitude occurs in the case of the antisymmetric A_4_ mode. Nevertheless, there is sufficiently large amplitude to excite the higher-order mode. Figure 6 illustrates spatial–temporal 2D FFT spectrum of the propagating modes, with red lines indicating symmetric modes and white lines indicating antisymmetric modes obtained by the SAFE method. The spatial–temporal 2D FFT spectrum of the measured A_1_ mode for the HDPE PE-300 sample (thickness of 20 mm), is presented in Figure 7a. The A_1_ mode contour obtained using the SAFE method is indicated by red line. The spectrum presented enables the recognition and identification of modes from Lamb waves propagating at various velocities. The extracted antisymmetric A_1_ mode, after filtering and amplitude normalization with respect to the maximum displacement, is presented in Figure 7b,c. It could be observed that the higher-order A_1_ mode can be excited from 30 kHz frequency. Theoretical and experimental investigations show the possibility of exciting the higher-order A_1_ mode with an ultrasonic low-frequency array or transducers.

The measurement results show that there is a normal component of antisymmetric higher-order A_1_ mode on the surface with sufficiently large amplitude, and it can be excited from a 30 kHz frequency, which indicates the possibility of exciting this mode through the air gap. The theoretical and measurement results show that it is possible to apply the lowest possible frequency and measure the higher-order antisymmetric mode A_1_. The red circle shows the selected lowest frequency of 42 kHz with maximum amplitude, presented in Figure 7c. This figure is intended to show the mode amplitude distribution in the frequency band and that it is possible to excite the A_1_ mode at a lower frequency, not only at 140 kHz. Higher-order A_1_ mode particle distributions were investigated using the SAFE method.

The particle displacement distributions across the sample of the antisymmetric mode A_1_ are given in Figure 8 and Figure 9. The presented results reveal that, in this mode, there are two main components: longitudinal (tangential) and vertical (normal to the surface). The considered normal component of the A_1_ has the same sign on both sides of the sample and relatively high amplitude on the surface. This observation leads to the assumption that at a frequency of 42 kHz, the A_1_ mode can be effectively excited with ultrasonic transducers or phased array.

In the next section, theoretical investigation using finite element modeling (FEM) is presented.

## 4. Theoretical Investigation Using Finite Element Modeling (FEM)

This chapter presents a numerical finite element (FEM) simulation focused on the propagation of guided wave modes in an HDPE PE-300 sample. The aim of this simulation was to excite different higher-order modes using a single-element ultrasonic transducer or a phased array. The ABAQUS software 7.00 package (Dassault Systemes, Johnston, RI, USA) was employed for simulation of the propagation of the appropriate guided wave modes in the HDPE PE-300 specimen. ABAQUS was selected for its high detail, the flexibility to configure materials and very accurate mesh control. By applying the 2D calculation method, higher-order modes in the plastic material were excited. The simulation diagram in Figure 10 illustrates the excitation of UGW by a single transducer. The excitation zone was at the distance *x* = 50 mm, away from the left edge of the sample. The excitation frequency was set at *f* = 140 kHz, with a burst of three periods. It is important to note that it is the same configuration, like that simulated by the SAFE method and measured by contact ultrasonic method with the ULTRALAB system.

To determine which mode (or modes) is excited by a single element of the array, one needs to estimate the velocities of the guided waves. This can be accomplished using two different methods: the zero-crossing method, which is extensively covered in our previous article [35], or another method, where phase velocities are typically determined from the distances ∆*d_gr_* and ∆*d_ph_* (*Y* coordinates axis), which were covered by the appropriate specific phase point during the appropriate time intervals ∆*t_gr_* and ∆*t_ph_* (*X* coordinate’s axis), Equations (3) and (4):(3)$$c_{ph}=\frac{\Delta d_{ph}}{\Delta t_{ph}},$$
(4)$$c_{gr}=\frac{\Delta d_{gr}}{\Delta t_{gr}}.$$

The numerical simulation results are presented in Figure 11 and Figure 12. In Figure 11, a simulated B-scan is presented. The zero-crossing instants of the normal displacements along the scanning line are presented in Figure 12a. The lines presented by different colors indicate different zero-crossing time instants. These results reveal that the A_0_ mode propagates with a phase velocity of *c_ph_* = 869 m/s, while the A_3_ mode has a velocity of *c_ph_* = 2542 m/s. The numerical simulation was performed at a frequency of 140 kHz. Please note that the A_3_ mode is excited at 140 kHz when the phase velocity is 2542 m/s, and it is presented in Figure 5, marked with a yellow circle. It can be concluded that sufficiently strong A_0_ mode is propagating with the phase velocity of *c_ph_* = 869 m/s and the group velocity of *c_gr_* = 910 m/s. Moreover, the A_3_ guided wave mode propagating with a phase velocity of *c_ph_* = 2512 m/s is visible in Figure 12c, after applying the zero-crossing method.

This paper also aims to show the feasibility of exciting a higher-order mode in the case of low-frequency excitation, but by applying an ultrasonic phased array. As a consequence, by changing the delay times, the same phased array can be applied to excite other higher-order modes. The identification of various higher-order waves and their application would facilitate the detection of various defects and delamination in materials, which could be applied in various industrial fields. Numerical investigation is presented to excite the first A_1_ higher-order wave using an ultrasonic linear array. The schematic diagram and simulation of the excitation of UGW by a linear array are depicted in Figure 13 and Figure 14. The array comprises eight strip-like elements (width of each, 1 mm). Theoretically, all spacings between array elements should be equal to *λ*_*A*1_ = 4.71 mm. In order to excite the A_1_ modes, the linear array elements are sequentially excited with the appropriate time delay required for this mode to propagate the distance from one element to another neighboring element. The delay time between each element (∆*τ* = 23.788 μs), was calculated using Equation (5):(5)$$\Delta \tau =\frac{\lambda_{A1}}{c_{Aph}\left(f,th\right)}$$
where *λ*_*A*1_ is the wavelength of the A_1_ mode, *f* = 42 kHz is the frequency, *th* = 20 mm is the thickness of the HDPE PE-300 sample. Each element excites the considered A_1_ mode, enhancing the amplitude of the propagating wave. The excitation signals with delay times (different colors) and B-scan of the HDPE PE-300 sample are presented in Figure 15a.

The ultrasound velocities of the modes shown in the B-scan were computed using Equation (3) and compared with the velocities of the UGW modes obtained through the SAFE method (Figure 5).

The phase velocity value estimated from the numerical simulation results (at 42 kHz) is *c_ph_* = 1948 m/s (Figure 15c). In comparison, the phase velocity of the A_1_ mode estimated by the SAFE method is *c_ph_* = 1980 m/s. It is possible to conclude that there is A_1_ mode propagating with the phase velocity *c_ph_* = 1980 m/s at 42 kHz. The numerical FEM results are very similar to those obtained from numerical simulations using the SAFE method.

## 5. Discussion

In the reviewed literature, many researchers have investigated composite or metallic specimens [22,24,25,28,29]. To date, there is little information on the investigated higher-order modes in HDPE samples. There are several works describing and applying ultrasonic waves to thermoplastic pipes [7,26,27]. In our case, the aim of this study was to efficiently excite and detect various higher-order guided Lamb wave modes in high-density polyethylene structures using single-element ultrasonic transducers and a phased array. Different methods are used to identify as many different higher-order modes as possible in the same specimen. This work studies the generation and reception of various higher-order UGW modes in a high-density polyethylene specimen using both contact ultrasonic transducers and a phased array. To excite various higher-order Lamb wave modes, selection of appropriate frequencies is crucial, depending on such factors as the mode type, the sample thickness, and material’s elastic properties. Therefore, initially, the properties of high-density polyethylene structures are outlined. Using these properties, the dispersion curves and higher-order UGW modes are estimated. The dispersion curves, calculated using the SAFE method, provide information about phase and group velocities of specific UGW modes for selected frequencies. Theoretical SAFE method calculations were compared with experimental contact measurements using ultrasonic transducers. Measurements were performed using contact transducers with a 0.5 mm diameter contact surface and operating at 50–350 kHz. Amplitudes of higher-order UGW modes were acquired using the ULTRALAB ultrasonic measurement system. A spatial–temporal 2D FFT spectrum analysis method was applied to obtain the corresponding velocities of the various modes. The experimental measurements were focused on exciting the A_3_ and A_4_ modes at a frequency of 140 kHz. Theoretical calculations using the SAFE method and experimental contact measurements exhibited good agreement. It is essential to note that, within the frequency range where higher-order UGW modes can propagate, multiple different modes can usually be excited. This can make it difficult to determine which modes have actually been excited. To address this, we propose a phase velocity *c_ph_* criterion for identification of the particular mode.

For practical purposes, the use of an ultrasonic array is more attractive. Therefore, another important objective of this work was to investigate the excitation of higher-order UGW modes using an ultrasonic array. The propagation of UGW modes in high-density polyethylene structures was investigated by FEM using the ABAQUS software package. The two different simulations were performed with a single-element ultrasonic transducer and an ultrasonic transducer array. The results of the SAFE method show that the higher-order A_1_ mode can be excited at a 42 kHz frequency. The ultrasonic phased array consists of eight elements, each with a width of 1 mm. Theoretically, distances between elements should be equal to *λ*_*A*1_ = 4.71 mm. To excite the A_1_ modes, the elements of the array are sequentially excited with a time delay necessary for the particular mode to propagate the distance between neighboring elements. The determined delay time between elements is ∆τ = 23.788 μs. The obtained simulation results confirm the excitation of the lowest higher-order UGW mode. The higher-order UGW A_1_ mode was identified from the particular phase velocity in the investigated 20 mm thick high-density polyethylene plate at an excitation frequency of 42 kHz. The phase velocity obtained from the numerical simulation results (frequency of 42 kHz) is *c_ph_* = 1948 m/s. According to the SAFE method, the phase velocity of the A_1_ mode was 1980 m/s. The velocity calculated by the FEM method agrees well with the velocity of the corresponding mode in the dispersion curves calculated by the SAFE method.

## 6. Conclusions

The purpose of this paper is to demonstrate the possibility of exciting higher-order modes using both single-element ultrasonic transducers and a phased array. The results obtained from both measurements and simulations clearly demonstrate the ability to efficiently excite and capture selected higher-order UGW modes with different phase velocities. It is important to note that, in the frequency range where higher-order guided waves modes can propagate, several different modes can usually be excited. To address this, we proposed a phase velocity *c_ph_* criterion for mode identification.

The simulation results are in good agreement with the measurement results. This shows that both methods can be used together. It was proved that, using an ultrasonic phased array, it is possible to excite the higher-order A_1_ mode.

In the near future, it is planned to carry out experimental investigations and simulations on composite samples in order to identify various defects present in such samples using the same low-frequency ultrasonic phased array.

## Figures and Tables

**Figure 1 materials-17-00163-f001:**
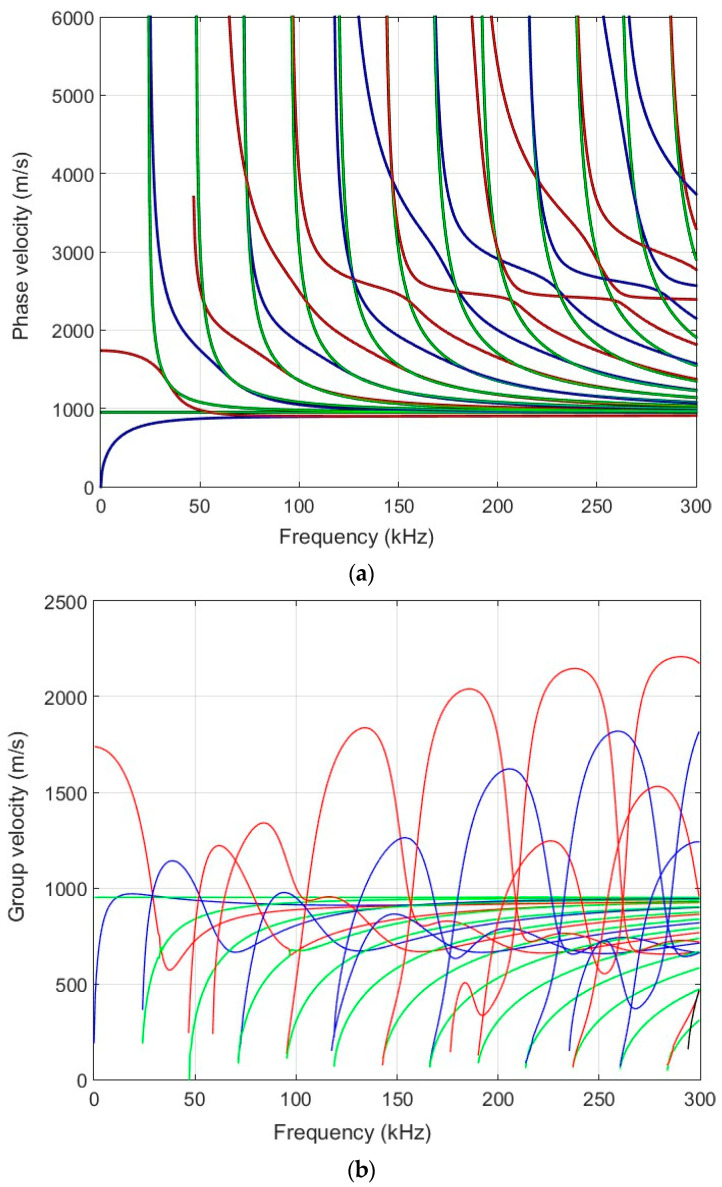
Calculated dispersion curves of the HDPE PE-300 sample (20 mm thickness): (**a**)—phase velocities; (**b**)—group velocities. Red solid lines—symmetric modes; blue—antisymmetric modes and green—shear modes.

**Figure 2 materials-17-00163-f002:**
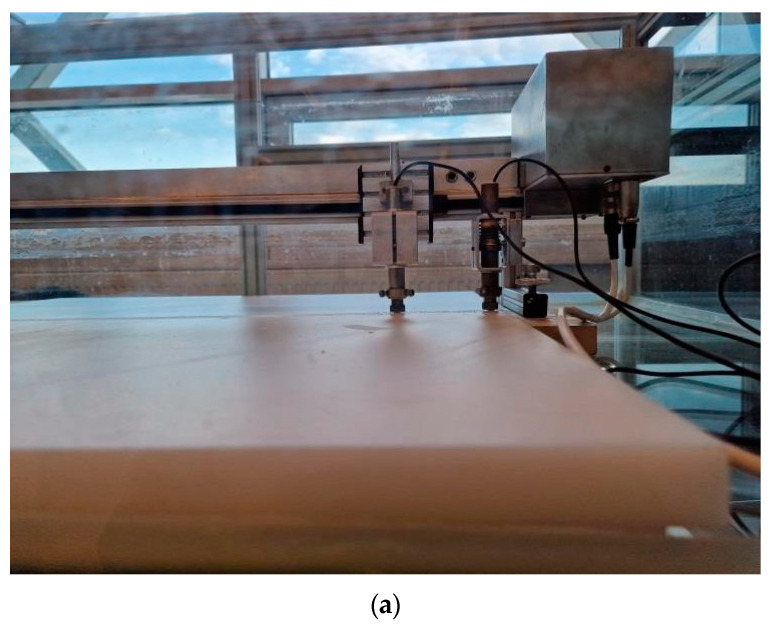
(**a**) Photo of the measurement set-up; (**b**) diagram of the measurement arrangement and linear scanning.

**Figure 3 materials-17-00163-f003:**
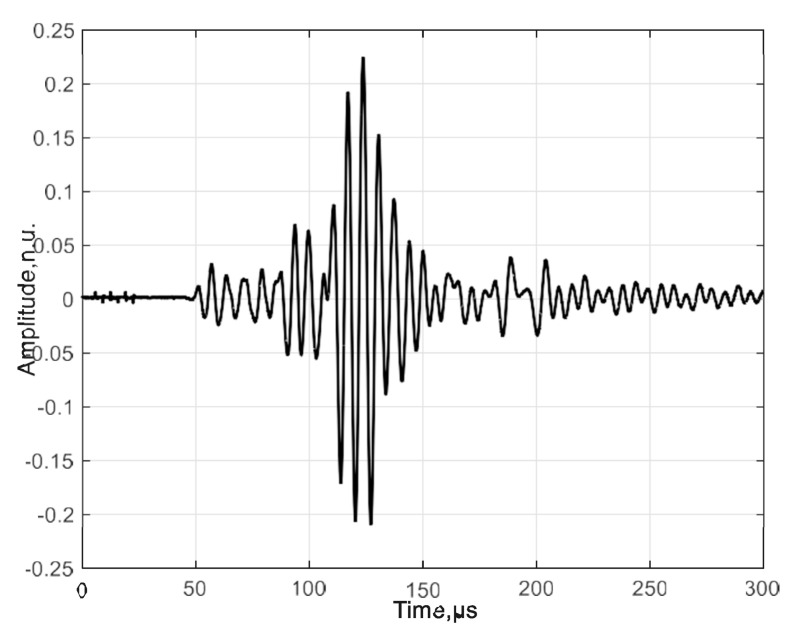
The received signal at the start position. The amplitude of the signal is presented in arbitrary units (n.u.).

**Figure 4 materials-17-00163-f004:**
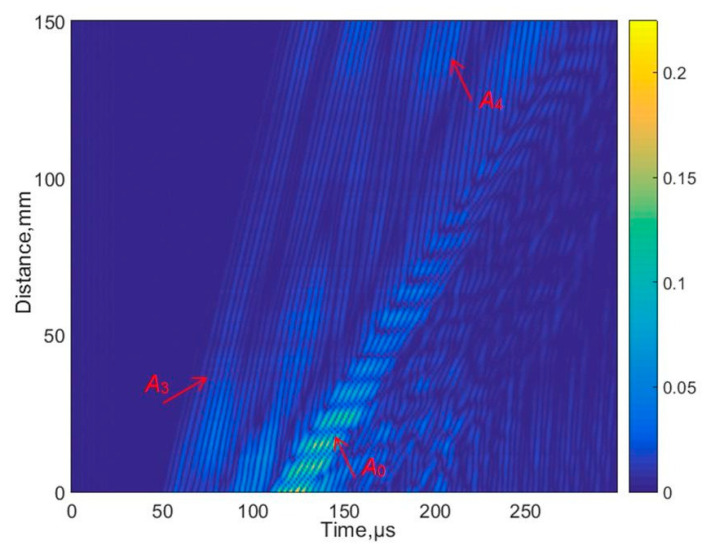
The measured B-scan image acquired in the case of the contact-type excitation and reception.

**Figure 5 materials-17-00163-f005:**
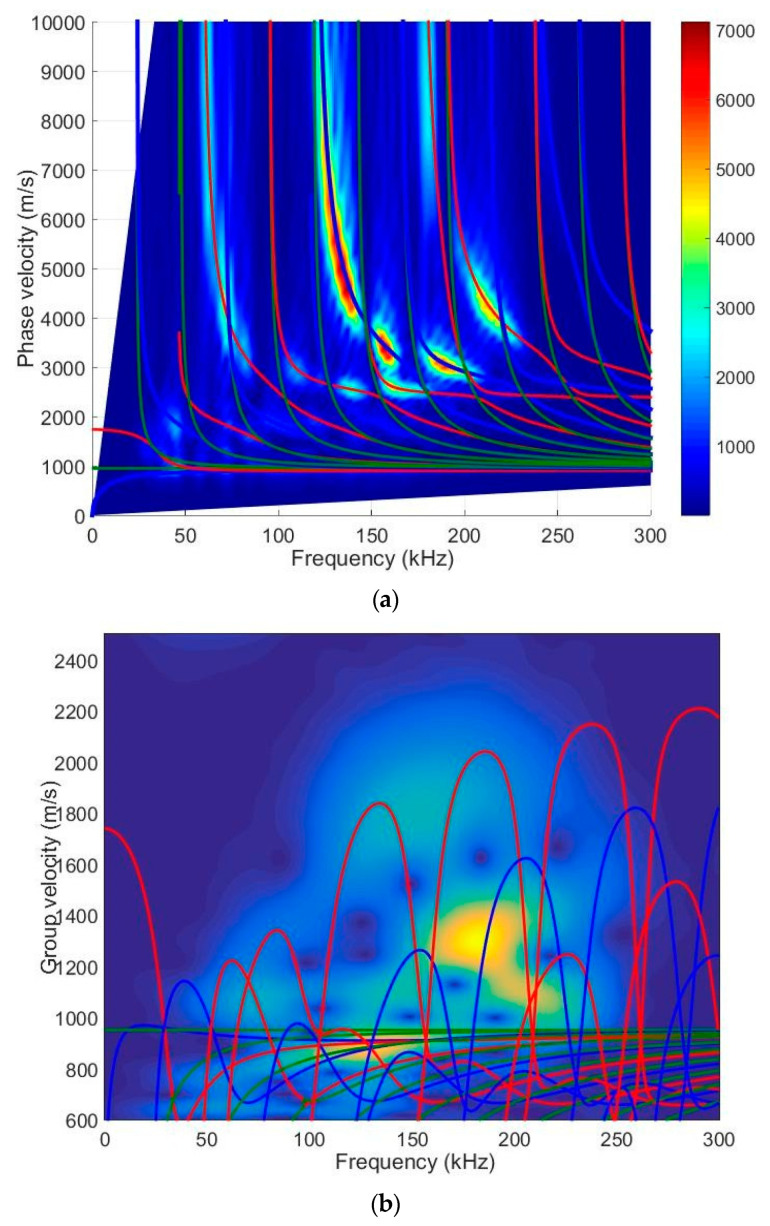
Two-dimensional FFT of the B-scan data and the dispersion curves obtained using SAFE method (blue lines—antisymmetric modes, red lines—symmetric modes and green line—shear modes): (**a**) presents all phase velocities of propagating modes; (**b**) presents all group velocities of propagating modes; (**c**) presents phase velocities of antisymmetric and symmetric modes up to 170 kHz.

**Figure 6 materials-17-00163-f006:**
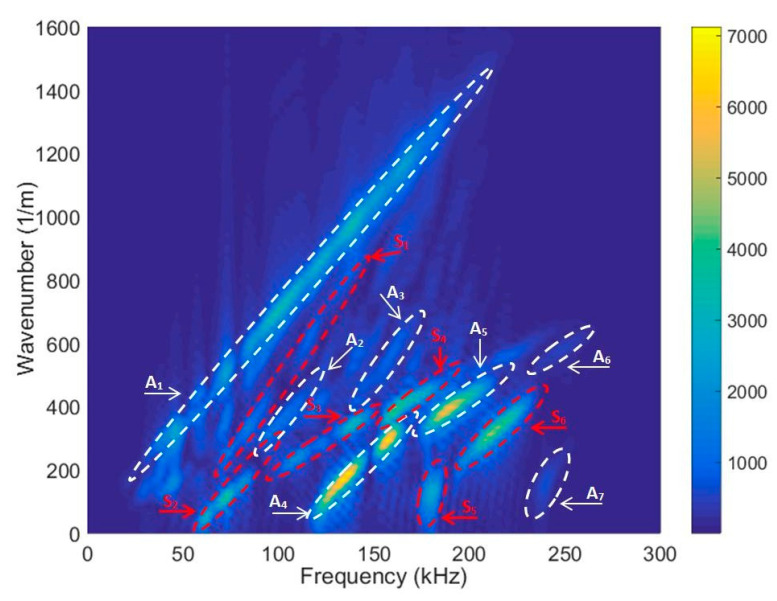
Spatial-temporal 2D FFT spectrum of the propagating modes. The red dotted lines are symmetric modes and white dotted lines—antisymmetric modes obtained by the SAFE method.

**Figure 7 materials-17-00163-f007:**
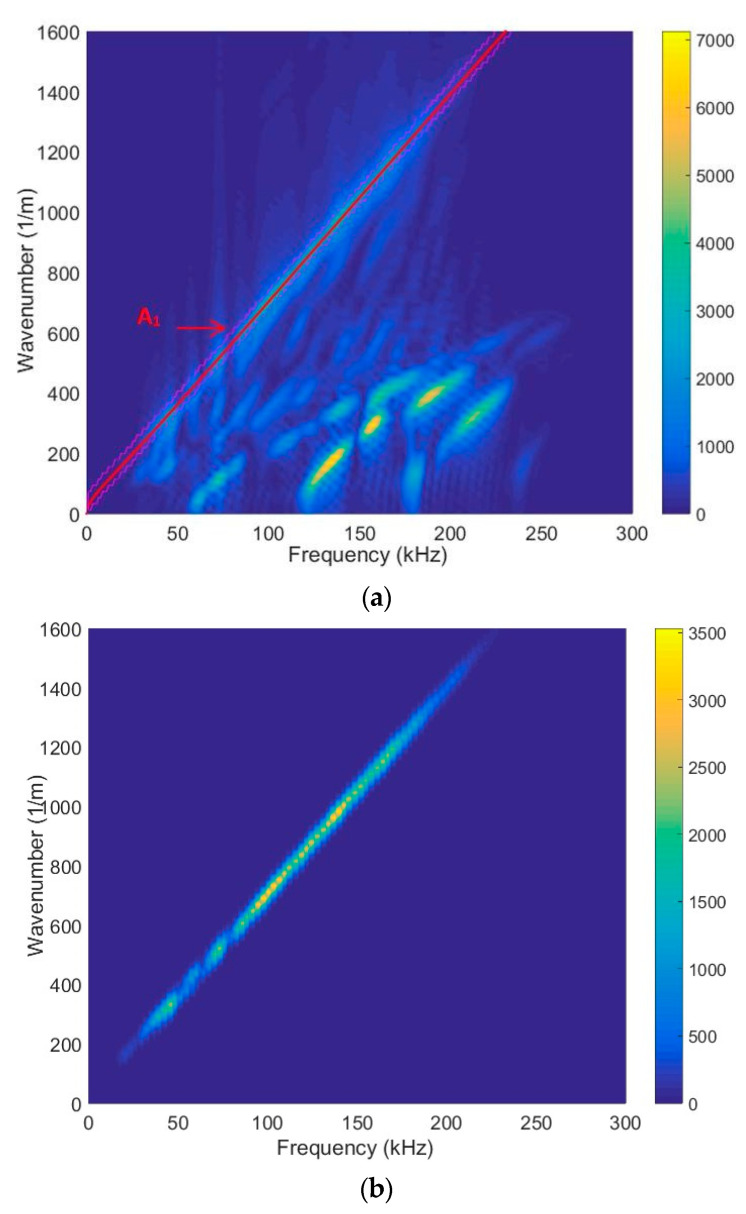
(**a**) Spatial–temporal 2D FFT spectrum of the measured A_1_ mode. The red line is the A_1_ mode contour obtained by SAFE; (**b**) filtered A_1_ mode; (**c**) amplitude normalized with respect to the maximum spectrum magnitude value of the filtered A_1_ mode. The red circle shows the selected lowest frequency of 42 kHz with maximum amplitude.

**Figure 8 materials-17-00163-f008:**
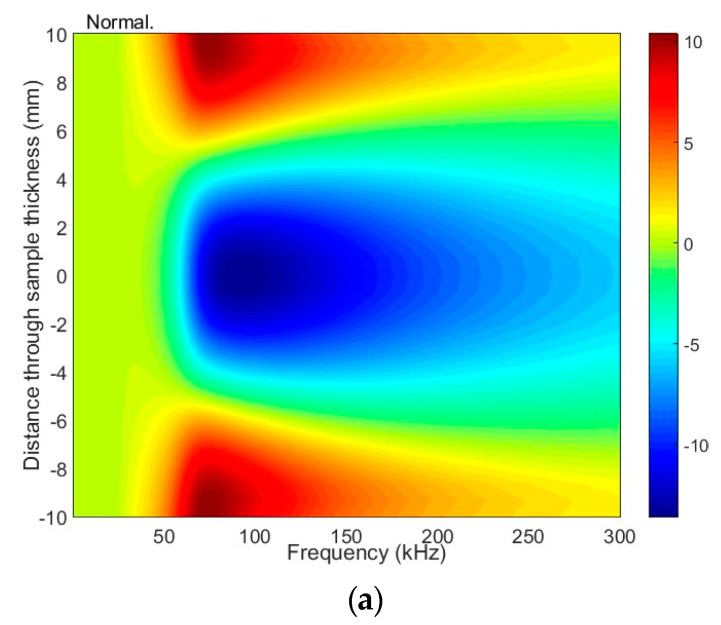
The distribution of particle displacement of the antisymmetric A_1_ mode across the sample (20 mm thickness) at excitation of 42 kHz: (**a**) frequency-dependent normal displacement; (**b**) frequency-dependent tangential displacement.

**Figure 9 materials-17-00163-f009:**
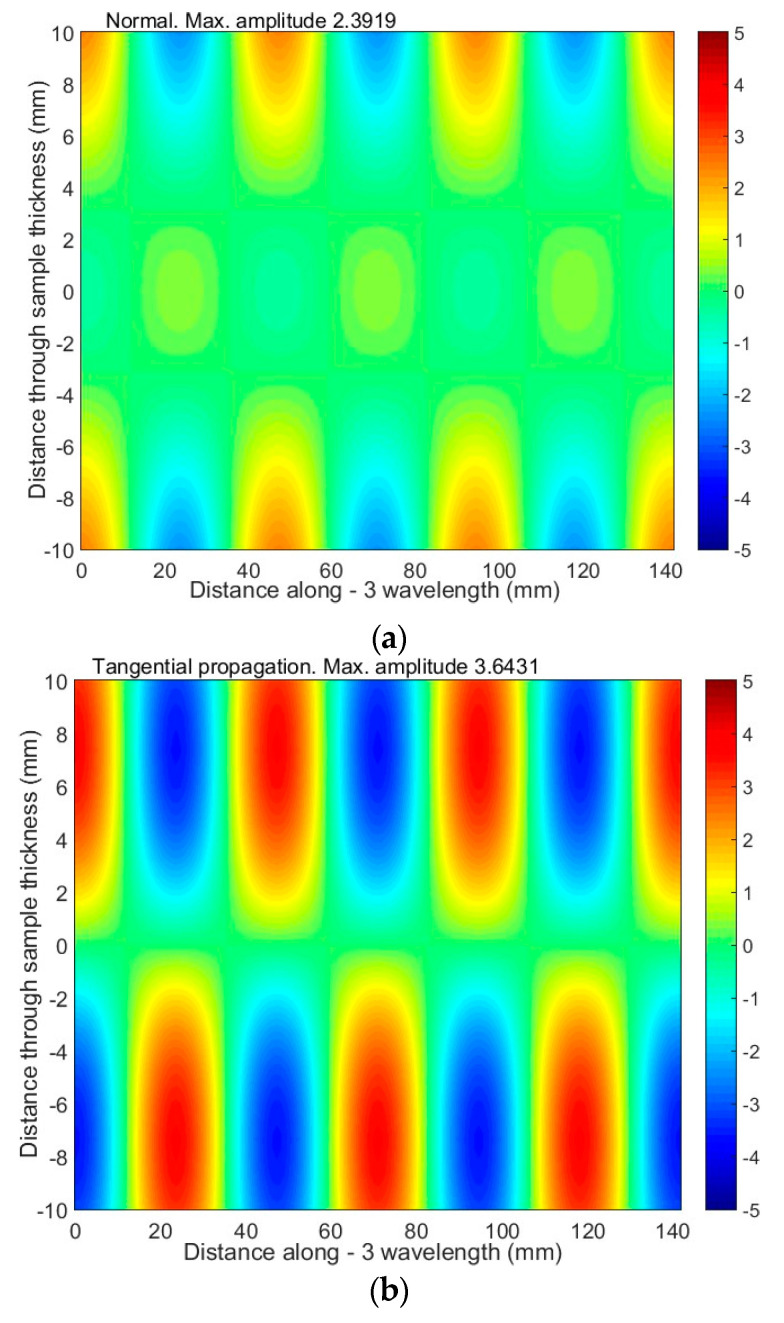
The distribution of particle displacement (normal and tangential) of the antisymmetric A_1_ mode across the sample (20 mm thickness) when excited at 42 kHz: (**a**,**b**) 3 wavelengths of the A_1_ mode are presented; (**c**) the red color is a tangential displacement and the blue color is a normal displacement.

**Figure 10 materials-17-00163-f010:**
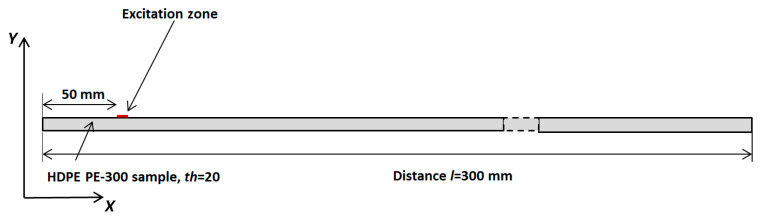
Simulation of the excitation of UGW using a single transducer.

**Figure 11 materials-17-00163-f011:**
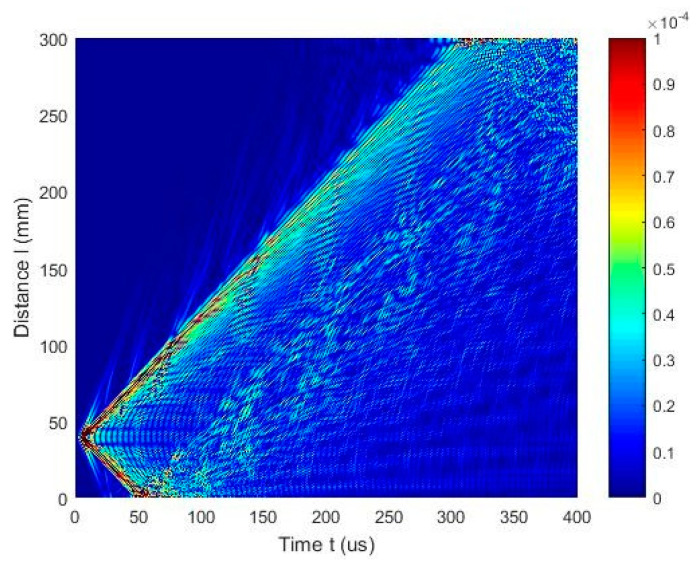
Simulated B-scan of the HDPE PE–300 sample.

**Figure 12 materials-17-00163-f012:**
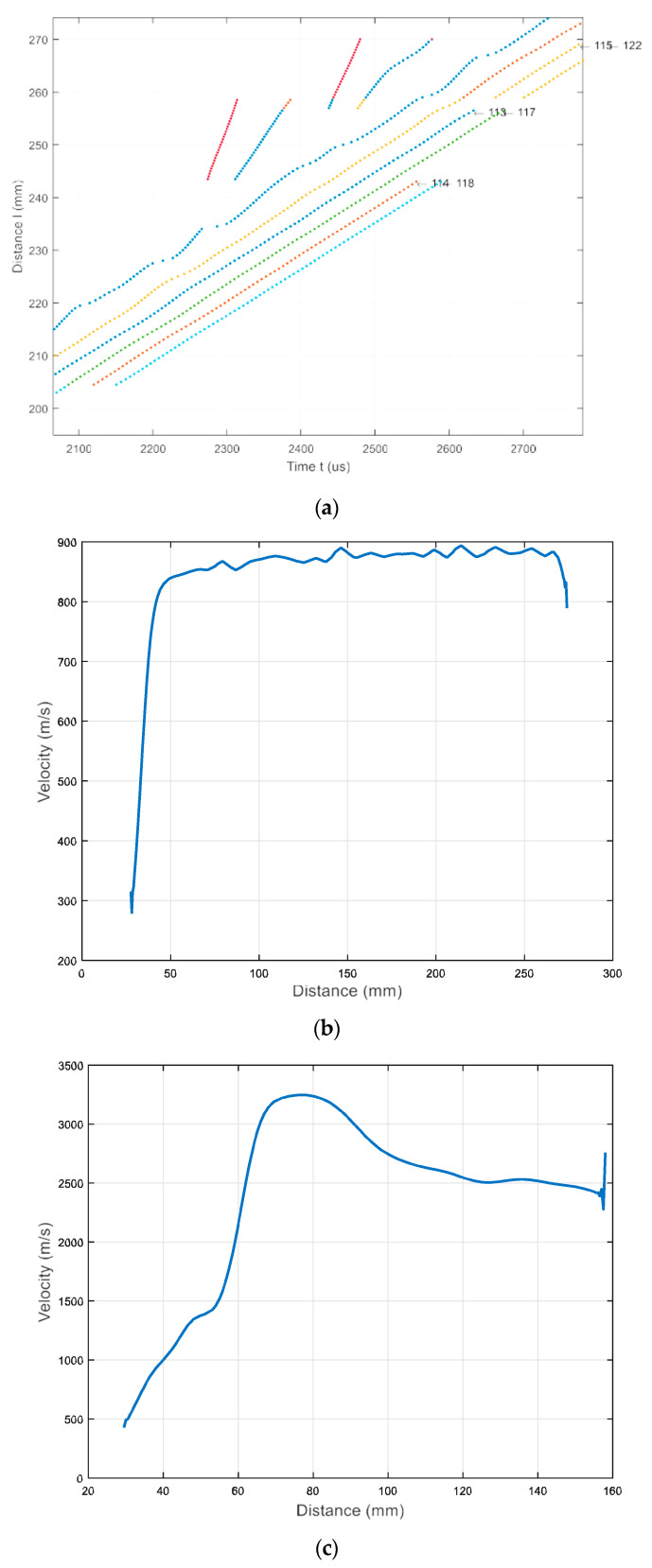
(**a**) Zero-crossing instants of the normal displacement signals acquired along the B-scan line; The lines presented by different colors indicate different zero-crossing time instants (**b**) mean value of the A_0_ mode velocity versus propagation distance X; (**c**) mean value of the A_3_ mode velocity versus propagation distance X.

**Figure 13 materials-17-00163-f013:**
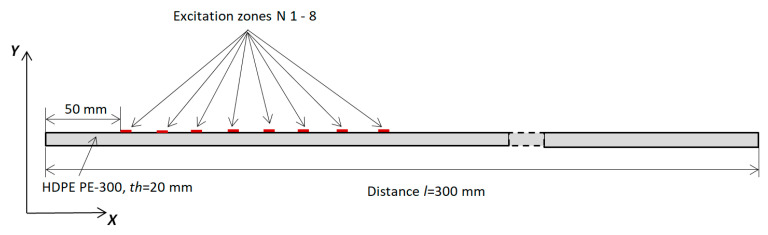
Diagram of the simulation model.

**Figure 14 materials-17-00163-f014:**
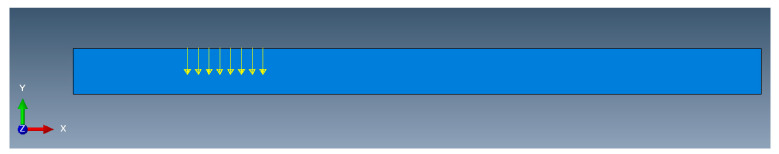
Simulation of UGW excitation by a linear array (yellow arrows represent excitation points).

**Figure 15 materials-17-00163-f015:**
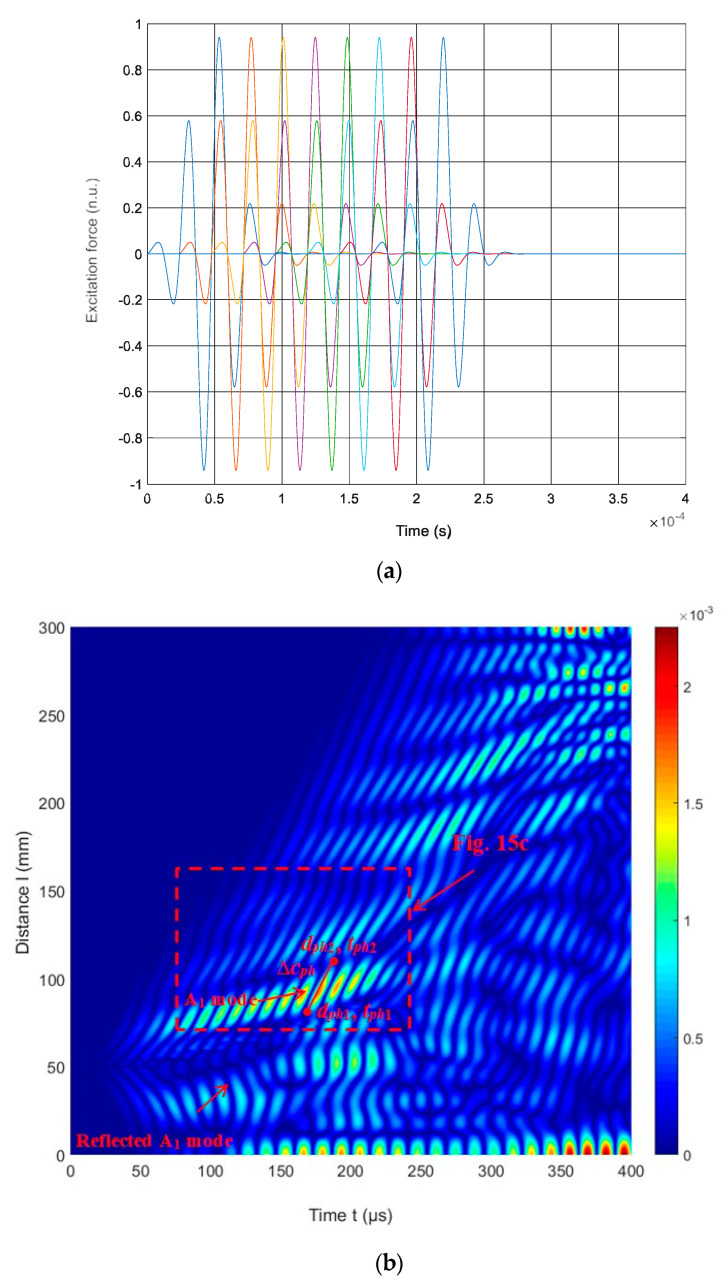
(**a**) Excitation signals with delay times (different colors); (**b**) the simulated B-scan of the HDPE PE-300 sample; (**c**) zoomed region of the B-scan in (**b**) (red rectangle).

**Table 1 materials-17-00163-t001:** The elastic parameters of the HDPE PE-300 sample.

Parameter	Value
*c* _11_	5.2 GPa
*c* _66_	0.85 GPa
Poisson’s coefficient, ν	0.4023
Density, *ρ*	941 kg/m^3^
Young’s modulus, E	2.3839 GPa

## Data Availability

Data are contained within the article.

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
