# Peer review of "Exciting and Detecting Higher-Order Guided Lamb Wave Modes in High-Density Polyethylene Structures Using Ultrasonic Methods"

_materials, 2023, doi:10.3390/ma17010163_

Round 1
Reviewer 1 Report
Comments and Suggestions for Authors
The authors successfully excite and receive various higher-order guided Lamb wave (UGW) modes in high-density polyethylene structures using both: ultrasonic single element transducers and a phased array. However, there are some key questions that the authors need to answer or solve.
1. The presentation of simulation data is missing in the abstract.
2. The logic of “Introduction” section should be improved.
3. The format of the references should be consistent and meet the requirements of the journal.
4. There are some writing errors in the text or figures.
Comments on the Quality of English Language
Language needs improvement. Please work through the manuscript carefully from this perspective.
Author Response
Dear Reviewer,
Thank you very much for your time and comments. We have taken your comments into account and have corrected and supplemented the text. We have provided the revised information below (attached file) and as well as in the text of the manuscript.
Best regards, authors

Reviewer 2 Report
Comments and Suggestions for Authors
The research article investigates high density polyethylene (HDPE), which is the most used thermoplastics on the market today, used in different industries, all over the world. The main objectives were to study and detect various higher-order guided Lamb wave modes in high-density polyethylene structures, employing single element ultrasonic transducers and phased array.
The article is well structured and well written, all the five sections describing the objectives, results and other findings in a scientific matter. The paper is divided into several sections, the first one is the one in which was performed the analysis over the HDPE samples with ultrasonic method, then the authors made a theoretical investigation of the HPDE samples using finite element modelling.
Although the article presents the results very clear, the authors could improve the conclusions and maybe add a few more research articles in the Introduction parts regarding other findings made by other authors.
The figures are well designed and the picture quality is high, easily readable and understandable.
Comments on the Quality of English LanguageThe quality of English is very good, only minor alterations are needed.
Author Response
Dear Reviewer,
Thank you very much for reviewing our manuscript and writing your review. We have noted your comments and have corrected the introduction and conclusions.
Best regards, authors

Reviewer 3 Report
Comments and Suggestions for Authors
The paper deals with the subject of exciting and detecting higher-order guided Lamb wave modes in high-density polyethylene structures using ultrasonic methods.
Please:
- verify the article linguistically,
- describe in detail the novelty of the article, in relation to other scientific papers,
- present measurement errors and repeatability of test results.
Comments on the Quality of English LanguagePlease verify the article linguistically.
Author Response
Dear Reviewer,
Thank you very much for your time and comments. We have taken your comments into account and have corrected and supplemented the text. The manuscript has been reviewed, read, and the inaccuracies found have been corrected. The following additional information, as well as additions and corrections, are included in the text.
Best regards, authors

Reviewer 4 Report
Comments and Suggestions for Authors
Dear Authors,
I attach comments in the file.
Yours sincerely,
Reviewer

Author Response
Dear Reviewer,
Thank you very much for your time and very detailed remarks. We have noted all your comments; and have corrected and supplemented the text. We have provided the revised information below (attached file), as well as in the text of the manuscript.
Best regards, authors

Round 2
Reviewer 4 Report
Comments and Suggestions for Authors
Dear Authors,
Please see the comments below.
Minor corrections are required.
1. 1. However, the authors should still complete the discussion of the results. Even if they wrote: “In the reviewed literature review, many researchers have investigated composite or metallic specimens.”, they did not provide any reference to literature.
2. 2. The point ”Author Contributions” should be still prepared in accordance with the requirements of the journal's Template.
After completing the discussion, I will accept the article for publication.
Yours sincerely,
Reviewer
Author Response
Dear Reviewer,
Yes, you are absolutely right, literary sources should have been indicated. We corrected it. Thank you very much for your consideration.
„Author Contributions " corrected and prepared according to the requirements of the journal template.
We sincerely thank you for your comments and recommendations. We will keep all of this in mind in the following articles.
Best regards, authors